# Association between Empagliflozin Use and Electrocardiographic Changes

**Daniel Antwi-Amoabeng** [1] , **Sunil Sathappan** [1] , **Bryce D. Beutler** [2,\*] , **Mark B. Ulanja** [3] , **Munadel Awad** [1] ,
**Nageshwara Gullapalli** [1] , **Phillip Duncan** [4] and **T. David Gbadebo** [5]

1   Department of Internal Medicine, Reno School of Medicine, University of Nevada, Reno, NV 89502, USA;
    antwiamoabeng@yahoo.com (D.A.-A.); ssathappan@med.unr.edu (S.S.); munadelll@gmail.com (M.A.);
    ngullapalli@med.unr.edu (N.G.)
2   Department of Radiology, Keck School of Medicine, University of Southern California,
    Los Angeles, CA 90033, USA
3   Department of Internal Medicine, Christus Ochsner Saint Patrick Hospital, Lake Charles, LA 70601, USA;
    markulanja@gmail.com
4   Division of Cardiology, Virginia Commonwealth University, Richmond, VA 23219, USA;
    phillip.duncan@vcuhealth.org
5   Department of Cardiology, Emory Decatur Hospital, Decatur, GA 30033, USA; tdgbadebo@gmail.com
\*   Correspondence: brycebeutler@hotmail.com

**Abstract:** Empagliflozin, a sodium-glucose transporter 2 inhibitor, has been shown to bind to late
sodium channels in mice cardiomyocytes. We sought to investigate the electrocardiographic (ECG)
features associated with empagliflozin use in patients with diabetes mellitus. We compared ECG
features of 101 patients before and after initiation of empagliflozin and found that empagliflozin was
associated with a significant increase in QRS duration among diabetes patients with heart failure.

**Keywords:** electrocardiography; empagliflozin; heart failure; SGLT2 inhibitors; late sodium channel

## 1. Introduction

The latest updates to the heart failure treatment guidelines by the American College
of Cardiology recommend the use of sodium-glucose cotransporter 2 (SGLT2) inhibitors,
either dapagliflozin and empagliflozin, in patients with heart failure with reduced ejection
fraction (HFrEF) irrespective of diabetes status as add-on therapy [1]. This recommendation
follows evidence from two clinical trials and a meta-analysis. First, the DAPA-HF trial
showed that in patients with HFrEF with or without diabetes, dapagliflozin reduced the
rate of worsening heart failure (HF) or cardiovascular (CV) death [2]. Second, in the
EMPEROR-Reduced trial, there was a significant reduction in the event rates of CV death
or HF hospitalizations in patients with diabetes treated with empagliflozin [3]. The effect
of these SGLT inhibitors was found to be consistent across the two trials in a subsequent
meta-analysis [4].

Recently, empagliflozin was shown to decrease the risk of CV death or HF hospitaliza-
tions in patients with HF with preserved or mid-range ejection fraction [5]. The European
Society of Cardiology also gave a Class I recommendation for the use of dapagliflozin,
empagliflozin, and sotagliflozin in patients with heart failure and diabetes to reduce HF
hospitalizations and CV death [6]. Although fatal arrhythmias were included in the adjudi-
cation of CV death in both the DAPA-HF and EMPEROR-Reduced trials, none reported the
incidence of non-fatal arrythmia as adverse effects or reduction in non-fatal arrhythmia
burden as an outcome. Per drug packet insert monographs, neither dapagliflozin nor
empagliflozin are associated with QTc prolongation in early clinical trials and effects on
ECG features associated with the drug have not been reported.

SGLT2 inhibitors have also gained attention for their potential antiarrhythmic effects,
including a significant reduction in the risk of atrial fibrillation, atrial flutter, and new-onset

arrhythmias [7–10]. The reported reduction in incidental arrhythmias from SGLT2 inhibitor use may be due to their ability to modify the comorbid risk factors for the development of those arrhythmias. However, the electrophysiologic mechanisms of SGLT2 inhibitors at the cellular level remain to be established. Recently, Philippaert et al. reported that empagliflozin targets late sodium channels and reduces late sodium current in murine models of heart failure [11]. The late sodium current is associated with increased calcium influx and prolongation of the plateau phase of the action potential [12]. Increased late sodium current occurs in heart failure, myocardial ischemia, and bradycardia, and has been associated with electrophysiologic and mechanical dysfunction [13].

SGLT2 inhibitors have the potential for wide adoption among prescribers given the clinical benefits. It is therefore important for clinicians to be aware of electrophysiologic effects of this class of medication. However, potential electrocardiographic changes associated with SGLT2 inhibitor use have not been reported. We hypothesized that empagliflozin (henceforth referred to as "Empa") would be associated with shortening of the myocardial action potential duration, which would manifest as a shorter QRS duration and QT interval on the surface electrocardiogram (ECG). We also postulated that Empa would cause slower ventricular rates due to inhibition of the sympathetic nervous system [14]. In this single-center cross-sectional study, we tested these hypotheses by reviewing the ECGs of patients prescribed Empa for management of diabetes mellitus.

## 2. Materials and Methods

We queried our healthcare system's database for encounters between 1 January 2020, and 31 December 2020, with at least one prescription renewal for empagliflozin and included patients who were at least 18 years old and had at least one ECG after initiation of Empa. We abstracted demographic data, comorbid conditions, all medication list before and after initiation of Empa, and cardiologist-confirmed ECG reports. For subjects who had before and after ECGs ($n = 78$), we used the most recent ECG obtained in the 12-month period prior to the initiation of Empa as the baseline ECG. We limited our sampling space to this window in order to limit the influence of disease progression and chronic medication changes in our observations. The frequency distribution of ECG rhythms before and after initiation of Empa was compared using Fisher's exact test of independence. The paired *t*-test was used to assess the difference between baseline and post-Empa ECG measurements. Medication lists before and after initiation of Empa were compared using the two-sample Z-test of equality of proportions. All tests were performed as 2-tailed and statistical significance levels were set at a $p < 0.05$ in all applied analysis. Statistical analysis was performed using Stata (StataCorp Release 16). The study was deemed exempt from institutional review board evaluation as it used existing patient data.

## 3. Results

Of the 361 patients who received at least one prescription for Empa during the study period, 101 had at least one ECG after initiation of Empa and were included in the study. Of these, 50.5% were female. The average age for the cohort was $65.7 \pm 10$ years at the time of first prescription and 48% were 65 years or older. Most of the patients (61.4%) were on the 10 mg daily dose; the remainder were on the 25 mg dose. Six patients (5.94%) died in the 18-month follow-up period. Hypertension and hyperlipidemia were the most common comorbid conditions. Table 1 displays a summary of the baseline comorbid conditions. There was no significant difference in the proportion of cardioactive medications prescribed before and after initiation of Empa (Table 2).

Seventy-eight patients had ECGs before and after initiation of Empa. Of these, 53.9% were female. Figure 1 illustrates the frequency distribution of ECG rhythms/findings at baseline and after initiation of Empa and the percent change in the observed frequencies. There was a decrease in the frequency of every arrhythmia type after the initiation of Empa, but there was no significant difference in the relative proportions of arrhythmias before and after medication initiation. Ventricular rate before initiation of Empa was faster compared

to the post-Empa ventricular rate (83 ± 13 versus 81 ± 15 beats per minute); however, this did not reach statistical significance (*p* = 0.65). An increase in several ECG metrics was observed after initiation of Empa, none of which were statistically significant: the PR interval increased by 4 milliseconds (ms) above baseline (170 ± 22 versus 174 ± 32 ms, *p* = 0.06); the QRS duration increased by an average of 6 ms (102 ± 25 versus 108 ± 38, *p* = 0.09); and the corrected QT interval (QTc) increased by 2 ms (454 ± 31 versus 456 ± 38, *p* = 0.65) among females but remained unchanged in males. Figure 2 illustrates the ECG measures before and after initiation of Empa at 10 milligrams (mg) and 25 mg. In a subgroup analysis involving 16 patients with heart failure, there was a significant increase in the QRS duration from 109 ± 22 ms to 120 ± 24 ms after initiation of Empa (*p* = 0.04). In a further subgroup analysis, the difference persisted only among the 10 patients who were on the 10 mg dose. Within this subgroup, the ventricular rate increased by 2 beats/min, PR interval increase by 2 ms, and the corrected QT interval increased by 3 ms. The increases were not statistically significant.

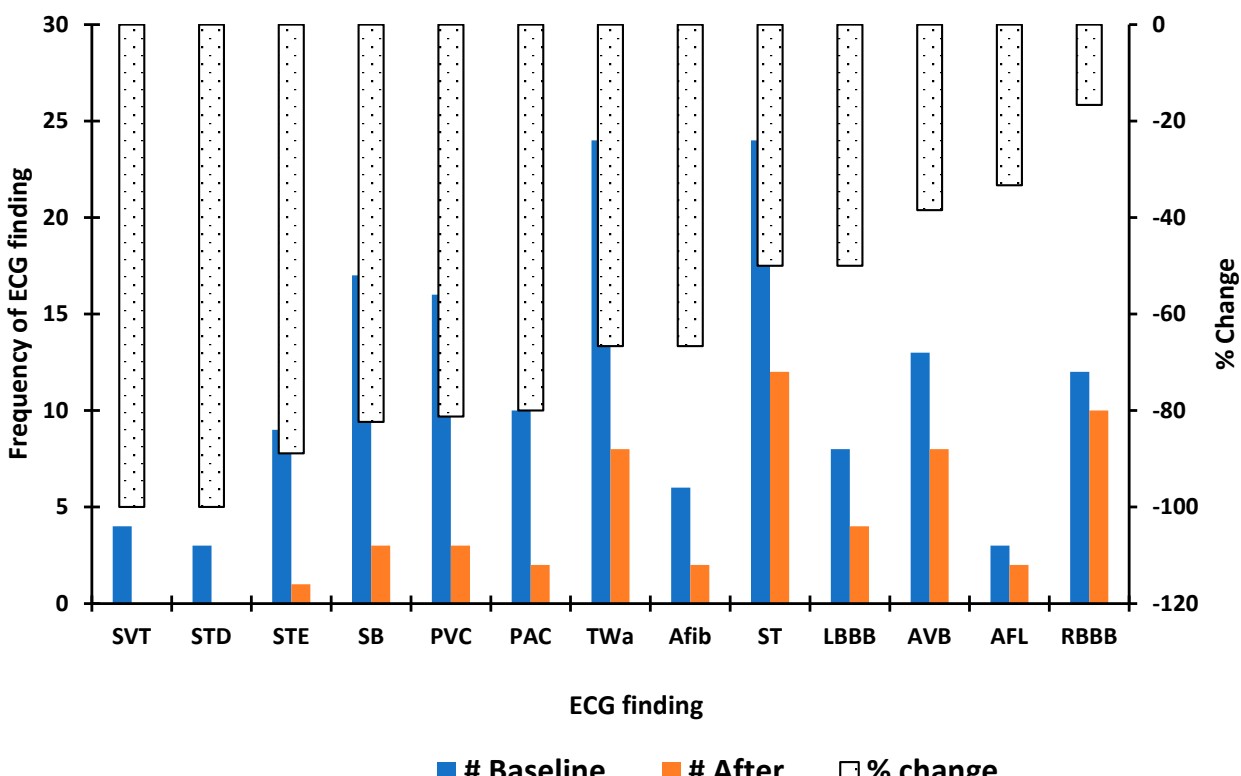

**Figure 1.** Paired frequency distribution and percent change in ECG rhythms before and after initiation of empagliflozin in 78 patients. ECG: electrocardiograpm; SVT: Supraventricular tachycardia; STD: ST segment depression; STE: ST segment elevation; SB: Sinus bradycardia; PVC: Premature ventricular contraction; TWa: T wave abnormalities; AFL: Atrial flutter; Afib: Atrial fibrillation; ST: Sinus tachycardia; LBBB: Left bundle branch block; AVB: Atrioventricular block; RBBB: Right bundle branch block; PAC: Premature atrial contraction.

**Table 1.** Summary of patient comorbid conditions.

| Condition | *n* (%) |
|---|---|
| Alcohol use | 18 (17.8) |
| Tobacco use | 61 (60.4) |
| Illicit drugs use | 12 (11.9) |
| Cancer | 25 (24.8) |
| Coronary artery disease | 33 (32.7) |
| Chronic obstructive pulmonary disease | 16 (15.8) |

**Table 1.** *Cont.*

| Condition | *n* (%) |
|---|---|
| Chronic kidney disease | 30 (29.7) |
| Heart failure | 17 (16.8) |
| Hypertension | 86 (85.2) |
| Hyperlipidemia | 89 (88.1) |
| Obesity | 57 (56.4) |
| Peripheral artery disease | 9 (8.9) |
| Sleep apnea | 25 (24.8) |
| Stroke | 13 (12.9) |

**Table 2.** Comparison of medication use before and after initiation of Empagliflozin.

| Medication | Empagliflozin Initiation | | *p*-Value |
|---|---|---|---|
| | before *n* (%) | after *n* (%) | |
| Aspirin | 26 (33.3) | 33 (42.3) | 0.48 |
| ACE inhibitors | 58 (74.4) | 44 (56.4) | 0.06 |
| Angiotensin receptor blockers | 26 (33.3) | 32 (41) | 0.53 |
| Beta blockers | 40 (51.3) | 44 (56.4) | 0.65 |
| Calcium channel blockers | 26 (33.3) | 25 (32.1) | 0.94 |
| Statin | 68 (87.2) | 66 (84.6) | 0.74 |

ACE: angiotensin-converting enzyme.

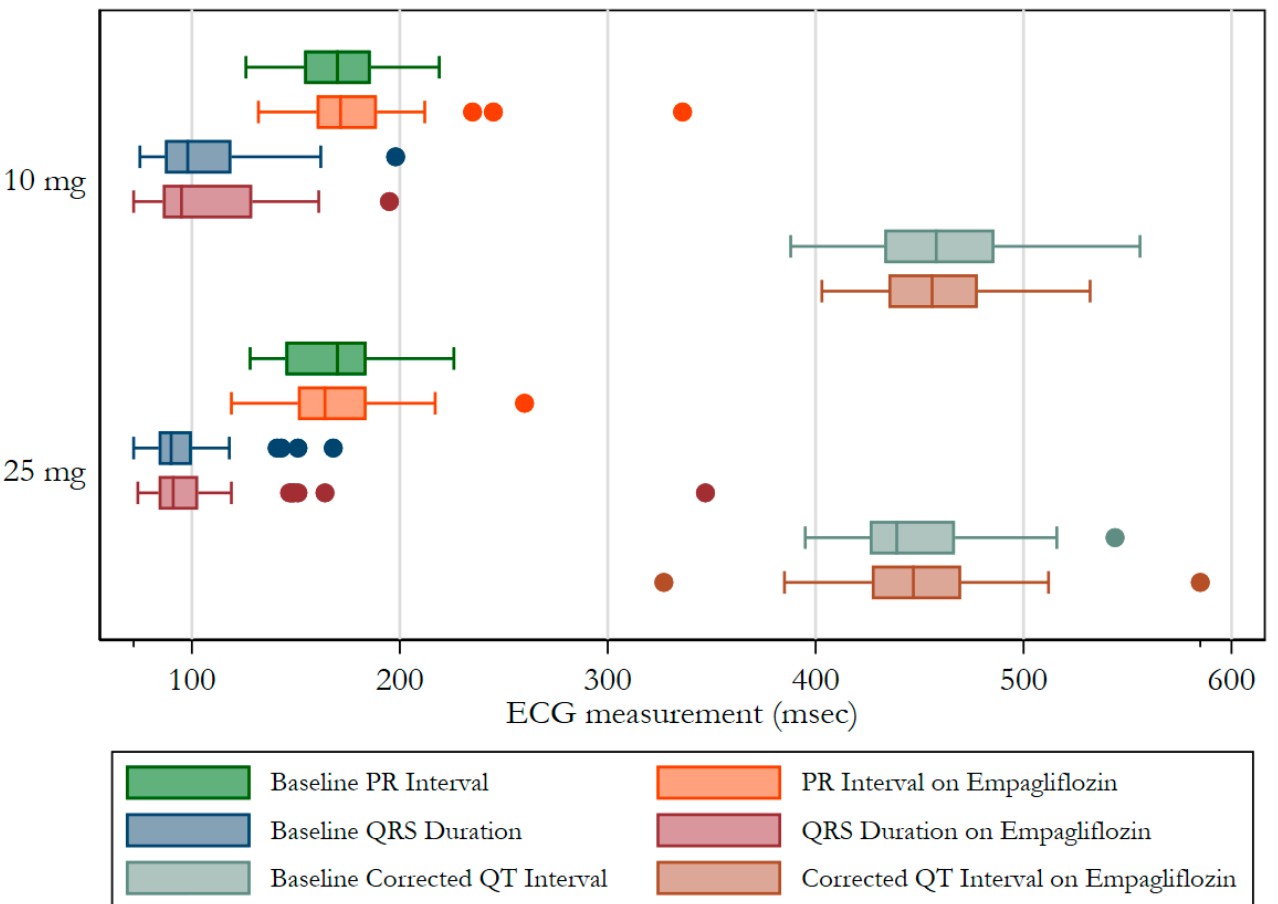

**Figure 2.** Comparison of ECG measurements before and after initiation of 10 mg versus 25 mg of empagliflozin. The colored circles indicate outliers.

## 4. Discussion

Pre-diabetes and diabetes mellitus (DM) are independent risk factors for the development of atrial fibrillation and ventricular arrhythmias [15–19]. The mechanism underlying DM-associated arrhythmias remains to be definitively established but is likely related to autonomic dysfunction, oxidative stress, and prolongation of the action potential duration [20–22]. Fortunately, most diabetes therapies, including metformin, thiazolidinediones, and dipeptidyl peptidase-4 inhibitors, have been shown to have a neutral or favorable effect on arrhythmia risk [23–25].

There is a paucity of data on the electrophysiologic effects of Empa and other SGLT inhibitors in patients with heart failure. Previous studies are limited to analysis of clinical event rates in clinical trial data or animal models. Post-hoc analysis of the DAPA-HF trial reported a 21% reduction in the risk of investigator-reported ventricular arrhythmia, resuscitated cardiac arrest, and/or sudden death in patients with HFrEF treated with dapagliflozin in addition to standard therapy [26]. In backward stepwise logistic regression analysis, dapagliflozin was associated with 20% reduction in the odds of the composite outcome (*p*-value was significant at <0.1). Although baseline QRS durations were reported, the authors did not include this in predictor models for the odds of the composite outcomes occurring. While investigator-reported events represent important clinical outcomes, the role of SGLT2 inhibitors in altering electrical or mechanical remodeling of the heart in HF remains to be elucidated. In mouse models of HF and metabolic syndrome, there was a significant shortening of the QT interval and effective refractory period, and fewer fibrotic areas in the ventricular tissue in the group treated with Empa [27]. Thus, Empa may have both clinically significant electrocardiographic and structural effects.

Inhibition of the late sodium current (late-$I_{Na}$) has a marked antiarrhythmic effect and pharmacologic agents targeting this pathway show a great deal of promise for patients with arrhythmogenic conditions [12,28–30]. The effect of antihyperglycemic agents on the action potential duration and late-$I_{Na}$ remains to be definitively established. However, Empa has recently been shown to inhibit late-$I_{Na}$ in murine models of DM and heart failure [11,31]. Prolonged late-$I_{Na}$ likely contributes to the arrhythmia burden in both conditions, and thus Empa has the potential to ameliorate both hyperglycemia and arrhythmias in patients with concomitant DM and heart failure [22,32]. The results of prior studies on the effect of Empa on cardiomyocytes of mice with heart failure and in diabetic rats suggest that the electrophysiologic manifestations of Empa treatment should be like those of known sodium channel inhibitors [11,31]. However, in our sample of 16 patients with diagnoses of both DM and heart failure, there was a significant increase in the QRS duration after initiation of Empa. Furthermore, there was a non-statistically significant increase in the corrected QT interval. We observed a reduction in the mean ventricular rate after initiation of Empa, but the difference did not reach statistical significance.

The principal finding of our study is a significant increase in the QRS duration among HF patients who received the 10 mg Empa dose from $109 \pm 22$ ms to $120 \pm 24$ ms. Although this observation was in a small number of patients, the finding could have significant clinical implications in HF patients. QRS prolongation may result in ventricular desynchrony, which can result in further compromised cardiac output, increasing myocardial demand and overall worsening HF symptoms [33,34]. Further, increased QRS duration is an independent risk factor for increased all-cause mortality and sudden death [35]. The 25 mg dose is recommended when more glycemic control is desired. We suggest that suboptimal glucose control in the 10 mg group may have resulted in a higher amount of sodium channel subunit glycosylation as compared to those receiving the 25 mg dose of Empa. Protein subunit glycosylation can occur non-enzymatically, and the extent of glycosylation is directly proportional to the glucose concentration of the milieu in which the protein exists. In patients with DM, there is persistent excess glycosylation and attaining serum glucose targets may alter the extent glycosylation [36]. Heavy glycosylation is a feature of many sodium channels and may affect the steady-state and function of these channels, the effect of which may differ in vivo compared to those in cell lines [37]. Furthermore,

heavy glycosylation may alter the docking domain and binding of Empa on the human cardiomyocyte late sodium channel as seen in other membrane proteins [38]. Empa use was associated with an insignificant reduction in the ventricular. Owing to its diuresis effects, it is expected that Empa use would correspond with reflex tachycardia, but this may be curtailed by the sympathetic attenuation effects of Empa [39].

Our analysis represents the first attempt to investigate electrophysiologic features of SGLT inhibitors in patients with coexistent DM and heart failure. Prior animal studies have been performed in models of either heart failure or diabetes mellitus; to the best of our knowledge, there are no data pertaining to the association between Empa and action potential duration or late-$I_{Na}$ in models representing coexistent DM and heart failure. It is therefore conceivable that there are interdependent effects that influence the response to Empa.

There are several limitations of this study inherent to its design. First, with only 101 patients in the cohort and 16 patients with both DM and heart failure, our findings should be interpreted cautiously. The generalizability of our findings is limited. Second, medication compliance and hemoglobin-A1C were not monitored throughout the duration of the study, thus effects of glycemic control on ECG changes could not be assessed. Third, other SGLTIs were not studied directly and the observed associations reported herein with Empa cannot be generalized to other agents within this class. Fourth, the study period is probably not long enough to assess the influence of duration of use on our findings. Lastly, owing to the small sample size, we were unable to control for confounders which could affect our findings. Despite these limitations, this study attempts to extrapolate proposed mechanisms of electrocardiographic effects in murine models to human subjects and represents a first step in translational research into the role, if any, of Empa on myocardial membrane physiology. Larger studies are warranted to definitively characterize electrophysiologic effects of Empa and other SGLT2Is in patients with heart failure and DM. A prospective cohort study in a carefully selected patient population could explore the relationship between Empa dose and electrocardiographic findings, assess whether the degree of glycosylation influences the association, and whether similar observations are seen in the other SGLT2Is.

**Author Contributions:** Conceptualization and writing—original draft, D.A.-A., S.S. and B.D.B. Methodology and data curation, M.B.U. and M.A. Supervision and writing—review and editing, N.G., P.D. and T.D.G. All authors have read and agreed to the published version of the manuscript.

**Funding:** This research received no external funding.

**Institutional Review Board Statement:** Ethical review and approval were waived for this study by the Institutional Review Board of the University of Nevada, Renno School of Medicine, as it used existing patient data.

**Informed Consent Statement:** Patient consent was waived, as all data were fully anonymized and summarized in aggregate.

**Data Availability Statement:** The data presented in this study are available on request from the corresponding author. The data are not publicly available, as all data were collected from a private healthcare system database.

**Conflicts of Interest:** The authors declare no conflict of interest.

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
