# Peer review of "Association between Empagliflozin Use and Electrocardiographic Changes"

_clinpract, doi:10.3390/clinpract12040059_

Round 1

Reviewer 1 Report

The topic treated in this paper is of interest and deserves in-depth analysis in light of the current relevance of the use of these drugs (SGLT2 inhibitors).

The Authors underline "………several limitations of this study inherent to its design………"; our perplexities actually concern the design of the study, but not only with regard to the number of cases studied but on the more properly methodological aspects. The available data derived from the database to which the authors referred, is to be considered inadequate in terms of cardiological items and profile.

In particular, the comorbidities prevalence is described in a very generic way, above all those of cardiological relevance; the prevalence of subjects affected by HF is certainly reduced (16.8%) without a further etiological definition, useful in the wide assessment of the studied population. In our opinion, the analyzed parameter must be contextualized in a multiparametric evaluation in order to put before the causal and concausal role of other parameters capable of correlating with the electrocardiographic changes.

Furthermore, a control group with similar clinical characteristics should have been defined within the statistical study.

Reviewer 2 Report

The manuscript "Association between empagliflozin use and electrocardiographic changes" compares ECG's before and after starting empagliflozin use in 78 patients, with the aim to detect changes hypothesized to be related to blocking the late sodium current in the repolarization phase by empagliflozin. The hypothesis is that such a block would decrease the action potential duration, because of decreased Na-Ca exchanging. The authors cannot confirm this hypothesis, as they find small average increases in PR and QRS durations. 

The study has major limitations in the design. The first is that there is no control group of similar patients not using empagliflozin (preferably matched for other medications and age). I may ask the authors what would be the natural course of this group of patients when they would not have received empagliflozin ? The second is that in the result there is a mention of a mean 18 months follow up, does that mean that the ECG's before and after empagliflozin were made with 18 months in between ?  There is no statement of timing of ECG's before and after empagliflozin in methods. For heart failure patients, especially those requiring new medications, there is a progressive disease, often also visible as increasing QRS duration. So, the study as presented cannot be used to demonstrate an effect of empagliflozin because there was no way to see that this was a result of empagliflozin, and because so much more could have happened to patients in the mean time. 

An interesting question though, this late sodium current inhibition. For ranolazine, also a late sodium current inhibitor, you would perhaps expect a decrease in QRS and QT time, but it was found that it could inhibit a rectifying potassium current, with a net effect of increasing the QRS and QT durations with 2 to 6 msec (Rayner-Hartley 2016). So, the finding that SGLT2 inhibition affects/ inhibits the late sodium current, does not mean that the overall effects of the medicine can be predicted. An other interesting aspect of sodium current inhibition is that it works as inhibition when the late sodium current is increased, for example in diabetes mellitus or heart failure as I read. I do not expect much if the sodium current is not activated. I would therefore select the ECG's at baseline showing an increased QRS and QT and find out what empagliflozin does on this 'activated' ECG's, compared to what it does on nonactivated ECG's. I would not select patients with atrial fibrillation. 

Also here, you need a right timing (1 week before and 1 week after starting SGLT2i), the right patients (preferably heart failure and SR), a parallel running matched group of patients not receiving empagliflozin or a cross over, you need a sample size calculation for the expected effect, and you need a proper study application with the medical ethics committee.     

Round 2

Reviewer 2 Report

The authors agree that the hypothesis to be tested (does empagliflozin change QRS or QT duration) cannot be tested by their study design. There is no control group, the duration between starting empagliflozin and measuring QRS is variable and long (12 months, mean 18 months), and several other influences could have played a role. It cannot be seen as a major limitation, it is impossible to present a conclusion. It is better that the authors improve the design of the study and start again. 

Author Response

We appreciate the additional comments from the reviewer.

Please note that this was an observational study and was not designed as a randomized controlled trial nor was it intended to definitively establish that SGLT2 inhibitors cause electrocardiographic (ECG) changes. The aim of our study was to determine if any ECG changes are observed among patients taking SGLT2 inhibitors, specifically empagliflozin, when monitored before and after initiation of treatment. A subset of the patients included in the study did demonstrate QRS prolongation when baseline ECGs were compared to ECGs obtained one year after beginning empagliflozin therapy, which was unexpected given the known mechanism of action of SGLT inhibitors. We agree that this does not indicate that empagliflozin causes ECG changes. However, it is an important observation that can serve as the foundation for further targeted research, which should indeed include a larger number of patients as well as a control group. We admit that our study is only hypothesis generating and is not intended to prove cause and effect; it should be evaluated as such.